# AN OPERATOR THEORETIC VIEW ON PRUNING DEEP NEURAL NETWORKS

**William T. Redman**
AIMdyn Inc.
UC Santa Barbara
wredman@ucsb.edu

**Maria Fonoberova & Ryan Mohr**
AIMdyn Inc.
{mfonoberova, mohrr}@aimdyn.com

**Ioannis G. Kevrekidis**
Johns Hopkins University
yannisk@jhu.edu

**Igor Mezić**
AIMdyn Inc.
UC Santa Barbara
mezic@ucsb.edu

## ABSTRACT

The discovery of sparse subnetworks that are able to perform as well as full models has found broad applied and theoretical interest. While many pruning methods have been developed to this end, the naïve approach of removing parameters based on their magnitude has been found to be as robust as more complex, state-of-the-art algorithms. The lack of theory behind magnitude pruning's success, especially pre-convergence, and its relation to other pruning methods, such as gradient based pruning, are outstanding open questions in the field that are in need of being addressed. We make use of recent advances in dynamical systems theory, namely Koopman operator theory, to define a new class of theoretically motivated pruning algorithms. We show that these algorithms can be equivalent to magnitude and gradient based pruning, unifying these seemingly disparate methods, and find that they can be used to shed light on magnitude pruning's performance during the early part of training.

## 1 INTRODUCTION

A surprising, but well-replicated, result in the study of deep neural network (DNN) optimization is that it is often possible to significantly "prune" the number of parameters after training with little effect on performance (Janowsky, 1989; Mozer & Smolensky, 1989a;b; LeCun et al., 1989; Karnin, 1990; Han et al., 2015; Blalock et al., 2020). Because DNNs have become increasingly large and require considerable storage, memory bandwidth, and computational resources (Han et al., 2015), finding a reduced form is desirable and has become an area of major research. In addition to the practical considerations, DNN pruning has become of theoretical interest, as the search for subnetworks that can be pruned with minimal effect on performance led to new insight on the success of large DNNs, via the development of the Lottery Ticket Hypothesis (LTH) (Frankle & Carbin, 2019; Frankle et al., 2020). A considerable body of literature has been devoted to expanding, interrogating, and critiquing the LTH, furthering the understanding of sparse DNNs.

While many different methods exist for choosing which parameters to prune, including ones that take into account the gradient and the Hessian of the loss function (LeCun et al., 1989; Hassibi & Stork, 1993; Yu et al., 2018; Lee et al., 2019; 2020), the simple approach of pruning based on a global threshold of parameters' magnitudes has been found to be robust and competitive with more complicated, state-of-the-art methods (Blalock et al., 2020). In addition, early work on the LTH found that magnitude pruning could be successfully performed early during training (although, iteratively pruning based on magnitude was found to yield considerably better results) (Frankle & Carbin, 2019). Subsequent research has related the point in training time when iterative magnitude pruning can be successfully applied to a phenomenon known as "linear mode connectivity" (Frankle et al., 2020). However, why magnitude pruning does not work before the emergence of linear mode

connectivity, is not well understood. Therefore, there is a need to develop new, theoretically driven tools for studying magnitude pruning, especially pre-convergence (Frankle et al., 2021).

DNN training can be viewed as a discrete dynamical system, evolving the associated parameters along a trajectory. Despite the fact that this perspective is familiar (indeed, it is often taught in introductions to gradient descent), the complex dependence of the trajectory on the choice of optimization algorithm, architecture, activation functions, and data has kept dynamical systems theory from providing much insight into the behavior of DNNs. However, the recent development of Koopman operator theory (Koopman, 1931; Koopman & Neumann, 1932; Mezić, 2005; Budišić et al., 2012), a rigorous, data-driven framework for studying nonlinear dynamical systems, has been successfully applied to DNNs and machine learning, more generally (Dietrich et al., 2020; Dogra & Redman, 2020; Tano et al., 2020; Manojlović et al., 2020; Mohr et al., 2021; Naiman & Azencot, 2021).

This recent success motivated us to ask whether we could leverage Koopman operator theory to provide insight into the pruning of DNNs. We first made use of Koopman methods to define a new class of pruning algorithms. One method in this class, which we refer to as Koopman magnitude pruning, is shown to be equivalent to magnitude pruning at convergence. Early during training, we again find that the two methods can be equivalent. The dynamical systems perspective afforded by Koopman pruning allows us to gain insight into the parameter dynamics present when the equivalence holds and when it does not. We find that the breaking of the equivalence occurs at a similar time that non-trivial structure emerges within the DNN. Finally, we extend our work beyond magnitude based pruning, showing that gradient pruning is equivalent to another form of Koopman pruning. Thus, the Koopman framework unifies magnitude and gradient pruning algorithms. Our contributions are the following:

- A new class of theoretically motivated pruning methods;
- New insight on how training dynamics may impact the success (or lack thereof) of magnitude pruning pre-convergence;
- A unifying framework for magnitude and gradient based pruning.

## 2 KOOPMAN OPERATOR THEORY BASED PRUNING

Koopman operator theory is a dynamical systems theory that lifts the nonlinear dynamics present in a finite dimensional state space, to an infinite dimensional function space, where the dynamics are linear (Koopman, 1931; Koopman & Neumann, 1932; Mezić, 2005). While gaining an infinite number of dimensions would appear to make the problem intractable, numerical methods have been developed over the past two decades to find good finite dimensional approximations of the relevant spectral objects (Mezić & Banaszuk, 2004; Mezić, 2005; Rowley et al., 2009; Schmid, 2010; Tu et al., 2014; Williams et al., 2015; Mezić, 2020). Given the powerful and interpretable tools that exist for studying linear systems, Koopman operator theory has found success in a number of applied settings. Below we discuss the basics of Koopman mode decomposition, a key element of Koopman operator theory, and use it to motivate a new class of pruning algorithms. We broadly refer to these methods as *Koopman based pruning*. For more details on Koopman operator theory, we refer the interested reader to Budišić et al. (2012).

### 2.1 KOOPMAN MODE DECOMPOSITION

The central object of interest in Koopman operator theory is the Koopman operator, $\mathbf{U}$, an infinite dimensional linear operator that describes the time evolution of observables (i.e. functions of the underlying state space variables) that live in the function space $\mathcal{F}$. That is, after $t > 0$ amount of time, which can be continuous or discrete, the value of the observable $g \in \mathcal{F}$, which can be a scalar or a vector valued function, is given by

$$\mathbf{U}^t g(p) = g\left[\mathbf{T}^t(p)\right]. \tag{1}$$

Here $\mathbf{T}$ is the nonlinear dynamical map evolving the system and $p$ is the initial condition or location in state space. In the case of training a DNN, $\mathbf{T}$ is determined by the choice of optimization algorithm and its associated hyperparameters (e.g. learning rate), architecture, activation functions, and order in which the training data is presented. If the optimizer is (a variant of) stochastic gradient descent,

then each epoch will have its own $\mathbf{T}$, since the dynamic map depends on the ordering of the training data (see Sec. 2.3). The DNN parameter values that are set at initialization, $\theta(0)$, are the associated initial condition [i.e. $p = \theta(0)$]. The observable, $g$, could be chosen to be the identity function or some other function that might be relevant to a given DNN, such as its loss or gradient. However, note that Eq. 1 must hold for all $g \in \mathcal{F}$. For the remainder of the paper, it will be assumed that the state space being considered is of finite dimension, and that $\mathcal{F}$ is the space of square-integrable functions.

The action of the Koopman operator on the observable $g$ can be decomposed as

$$\mathbf{U}g(p) = \sum_{i=0}^{R} \lambda_i \phi_i(p)\mathbf{v}_i, \tag{2}$$

where the $\phi_i$ are eigenfunctions of $\mathbf{U}$, with $\lambda_i \in \mathbb{C}$ as their eigenvalues and $\mathbf{v}_i$ as their Koopman modes (Mezić, 2005). The decomposition of Eq. 2 is called the Koopman mode decomposition.

The Koopman mode decomposition is powerful because, for a discrete dynamical system, the value of $g$ at $t \in \mathbb{N}$ time steps in the future is given by

$$g\left[\mathbf{T}^t(p)\right] = \mathbf{U}^t g(p) = \sum_{i=0}^{R} \lambda_i^t \phi_i(p)\mathbf{v}_i. \tag{3}$$

From Eq. 3, we see that the dynamics of the system in the directions $\mathbf{v}_i$, scaled by $\phi_i(p)$, are given by the magnitude of the corresponding $\lambda_i$. Assuming that $|\lambda_i| \leq 1$ for all $i$, finding the long time behavior of $g$ amounts to considering only the $\phi_i(p)\mathbf{v}_i$ whose $\lambda_i \approx 1$.

Given that the Koopman operator is linear, and it is known that even simple DNNs can have parameters that exhibit nonlinear dynamics (Saxe et al., 2014), it may seem that applying Koopman methods to the training of DNNs is an over simplification. However, the Hartman-Grobman Theorem, a foundational result in dynamical systems theory, guarantees local neighborhoods around hyperbolic fixed points where the linearization captures the dynamical properties of the full, nonlinear system (i.e. the two are conjugate) (Wiggins, 2003). Additional work by Lan & Mezić (2013) extended the neighborhood where this conjugacy holds to the entirety of the basin. Therefore, as long as the DNN is in a basin of attraction of some fixed point, we can assume that the Koopman mode decomposition will provide an accurate representation to the true, nonlinear dynamics.

There exist a number of different methods for computing the Koopman mode decomposition. While they differ in their assumptions on the dynamics of the system, they are all data-driven and require multiple snapshots of some $g$. In the case of DNN training, we can take $g$ to be the identity function, and collect $\tau + 1$ column vectors of the values the DNN parameters, $\theta$, each taken at a different training step. That is, to compute the Koopman mode decomposition, the data matrix $\mathbf{D} = [\theta(0), \theta(1), ..., \theta(\tau)] \in \mathbb{R}^{N \times (\tau+1)}$ must be constructed. For more details on the method used in this paper to compute the Koopman modes, and its computational demands, see Appendix A.

## 2.2 KOOPMAN BASED PRUNING

Any combination of DNN optimizer, architecture, activation function, and ordering of training data has an associated Koopman operator, $\mathbf{U}$, which governs the flow of DNN parameters during training. This Koopman operator, in turn, has various directions along which the flow exponentially grows and shrinks (i.e. the $\phi_i[\theta(0)]\mathbf{v}_i$).

Previous work has shown that, over a non-trivial window of training, approximations of the Koopman operator can well capture weight and bias dynamics (Dogra & Redman, 2020; Tano et al., 2020), suggesting that the $\phi_i[\theta(0)]\mathbf{v}_i$ may be useful for pruning. If we order the modes by the real part of their eigenvalues, i.e. $\text{re}(\lambda_1) > \text{re}(\lambda_2) > ... > \text{re}(\lambda_R)$, and assume that $\lambda_1 = 1$ (a reasonable assumption in the case of stable dynamical systems, which we find corroborated in our experiments below), then $\phi_1[\theta(0)]\mathbf{v}_1$ corresponds to the predicted fixed point of training. That is, the first mode identifies the parameter values that the DNN would take after training reached convergence. All additional modes would identify parameters that experienced increasingly larger exponential decays.

If $\phi_1[\theta(0)]\mathbf{v}_1$ is accurate in its approximation of the fixed point, then pruning parameters based on their magnitude in $\phi_1[\theta(0)]\mathbf{v}_1$ could prove to be beneficial. We refer to this as *Koopman magnitude*

---

**Algorithm 1** General form of Koopman based pruning.

1. Construct the data matrix $\mathbf{D} = [\theta(0), \theta(1), ..., \theta(\tau)]$ from $\tau + 1$ snapshots of the parameter values $\theta$ during training.

2. Compute the Koopman mode decomposition from $\mathbf{D}$ to obtain $R$ Koopman triplets, $(\lambda_i, \phi_i, \mathbf{v}_i)$.

3. Use a scoring function, $s$, that is a function of $(\lambda_i, \phi_i, \mathbf{v}_i)$, to create a pruning mask, $m$, that compresses the network by an amount $c$. In the case of KMP, $s(\theta, \lambda_i, \phi_i, \mathbf{v}_i) = |\phi_1[\theta(0)]\mathbf{v}_1|$, assuming that $\lambda_1 = 1$.

4. Create the pruned DNN, $f(x; \theta \odot m)$.

---

*pruning* (KMP) to denote its similarity to global magnitude pruning (GMP), a popular and robust approach to pruning (Blalock et al., 2020). We make this connection explicit in Section 3.

Alternatively, the exponentially decaying modes, i.e. $\phi_2[\theta(0)]\mathbf{v}_2, \phi_3[\theta(0)]\mathbf{v}_3...$, might be expected to convey information about which parameters are most coherently dynamic. We refer to the strategy that prunes parameters based on their magnitude in these modes as *Koopman gradient pruning* (KGP) to denote its similarity to gradient pruning. We make this connection explicit in Section 5.

These algorithms, KMP and KGP, are two of many possible realizations of pruning based on the Koopman mode decomposition. For example, KMP can be easily extended to layer magnitude pruning, by splitting $\phi_1[\theta(0)]\mathbf{v}_1$ into subvectors and pruning each layer based on thresholds computed from each subvector. Algorithm 1 provides a general form for Koopman based pruning. Note that we follow standard practice in the pruning literature and denote a DNN by $f(x; \theta)$, where $x$ is its input and $\theta$ is its $N$ parameters. The goal of pruning is to find a mask, $m \in \{0, 1\}^N$, such that the accuracy of $f(x; \theta \odot m)$ is comparable to the original DNN. Here $\odot$ denotes element-wise multiplication. To determine which parameters to prune (i.e. which $m_i = 0$), a scoring function, $s$, is used. In the case of GMP, $s(\theta_i) = |\theta_i|$. The DNN is compressed an amount $c$ by keeping only the $\frac{100}{c}\%$ of parameters with the largest scores. If $c = 4$, then only the parameters with scoring function values that are in the top 25% are kept.

### 2.3 KOOPMAN OPERATOR'S DEPENDENCE ON ORDER OF TRAINING DATA

As noted above, the dynamical map involved in training a DNN, $\mathbf{T}$, is dependent on the choice of optimizer, architecture, activation function, and order in which the training data is presented. Given that the action of the Koopman operator is related to $\mathbf{T}$ via Eq. 1, the Koopman mode decomposition must also be dependent on these quantities. In the experiments that are performed in the remainder of this paper, optimizer, activation function, and architecture will be fixed ahead of time. However, because we made use of SGD, the ordering of the training data, referred to in some literature as "SGD noise" (Frankle et al., 2020), will vary epoch to epoch.

A consequence of this is that, at each epoch, the dynamical system being considered will change, as the mapping from $\theta(t)$ to $\theta(t + 1)$ will depend on which subset of training data is shown at training iteration $t$. Note that it will also depend on the batch size (i.e. what fraction of the training data is shown before updating $\theta$), however this will also be fixed in the experiments we perform. While Koopman operator theory has been extended to random and non-autonomous dynamical systems (Maćešić et al., 2018; Črnjarić-Žic et al., 2020), we can view the training over the course of an epoch as a deterministic process, by fixing ahead of time the ordering of the training data. For the remainder of this paper, we will be computing the Koopman mode decomposition for each epoch by using only the parameter evolution data from that epoch.

The impact that SGD noise has on a DNN has been found to depend on the amount of training (Frankle et al., 2020). Near initialization, two copies of the same DNN trained on different orderings of the data lead to convergence on different local minima, whose linear path sees a "hill" (i.e. increase) of testing error. This suggests that they are in different parts of the loss landscape. However, with increased training, SGD noise brings the two copies of the DNN to the same "valley", where their linear path has no significant rise in testing error. This phenomenon is called linear mode connectivity (Frankle et al., 2020), and it is known to be related to the success of being able to find

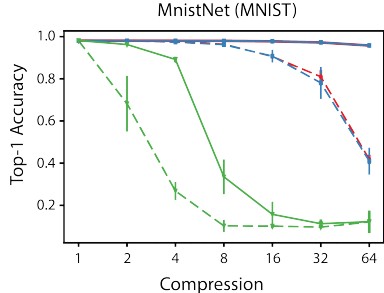 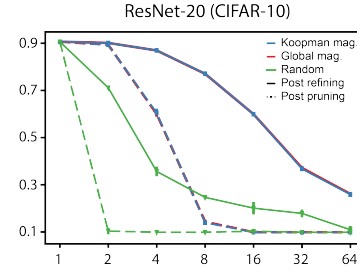

Figure 1: **Equivalence of Koopman and global magnitude pruning at convergence.** Left, compression vs. accuracy of MnistNet trained on MNIST. Right, compression vs. accuracy of ResNet-20 trained on CIFAR-10. Dashed lines are mean accuracy immediately after pruning and solid lines are mean accuracy after a single epoch of refining. Error bars are standard deviation.

winning tickets in the Lottery Ticket Hypothesis framework. However, the cause of its emergence, and whether it is related to other dynamical phenomena, are not currently well known.

## 3    EQUIVALENCE OF GLOBAL AND KOOPMAN MAGNITUDE PRUNING IN THE LONG TRAINING LIMIT ($t \to \infty$)

We start by examining how global and Koopman magnitude pruning compare in the long-training time limit. We consider a DNN, initialized with parameters $\theta(0)$, optimized via SGD on training data $x$, with the order of the $i^{\text{th}}$ epoch's data denoted by $x_i$. As the number of epochs goes to $\infty$, the DNN will converge to $\theta^*$, a local minimum in the loss landscape. This requires that $||\theta(t) - \theta^*||_2 \to 0$, as $t \to \infty$. Therefore, for $\epsilon \in \mathbb{R}^+$, there exists $t_\epsilon \in \mathbb{N}$, such that $||\theta(t) - \theta^*||_2 < \epsilon$, for all $t > t_\epsilon$. Here, $|| \cdot ||_2$ is the $l^2$ norm. Note that this additionally implies that at $t_\epsilon$ the DNN is in the basin of attraction of $\theta^*$.

Previous work on approximating the Koopman operator has found that if a system is sufficiently near a fixed point (and if the trajectory is sufficiently "nice"), then the computed Koopman mode decomposition should well approximate the true dynamics. That is, for $\epsilon \in \mathbb{R}^+$, there exists $t_\epsilon^{\text{Koop}} \in \mathbb{N}$, such that $||\tilde{\theta}^* - \theta^*||_2 < \epsilon$, for all $t > t_\epsilon^{\text{Koop}}$. Here, $\tilde{\theta}^* = \phi_1[\theta(t_\epsilon^{\text{Koop}})]\mathbf{v}_1$ is the computed Koopman mode corresponding to $\lambda = 1$.

Assuming that a DNN has trained for at least $\max(t_\epsilon, t_\epsilon^{\text{Koop}})$ training steps, then

$$||\tilde{\theta}^* - \theta(t)||_2 < 2\epsilon, \tag{4}$$

If $\epsilon$ is sufficiently small, then $m(|\tilde{\theta}^*|, c) \approx m(|\theta(t)|, c)$, and the two methods are thus equivalent.

To test whether this result holds in practice, we used ShrinkBench, a recent open source toolkit for standardizing pruning results and avoiding bad practices that have been identified in the pruning literature (Blalock et al., 2020). The results of pruning a custom ShrinkBench convolutional DNN called "MnistNet" (Blalock et al., 2020), pre-trained on MNIST [1], and ResNet-20, pre-trained on CIFAR-10 [2], are presented in Fig. 1. Each DNN was trained from a point near convergence for a single epoch, so that the parameter trajectories could be captured. All hyperparameters of the DNNs were the same as the off-the-shelf implementation of ShrinkBench, except that we allowed for pruning of the classifier layer. Training experiments were repeated independently three times, each with a different random seed. Our code has been made publicly available [3].

We found that, for both MnistNet and ResNet-20, KMP and GMP produced nearly identical results, both immediately after pruning and after one epoch of refinement (Fig. 1). These results support the claim developed above that KMP and GMP are equivalent in the long training time limit.

---

[1]https://github.com/JJGO/shrinkbench-models/tree/master/mnist
[2]https://github.com/JJGO/shrinkbench-models/tree/master/cifar10
[3]https://github.com/william-redman/Koopman_pruning

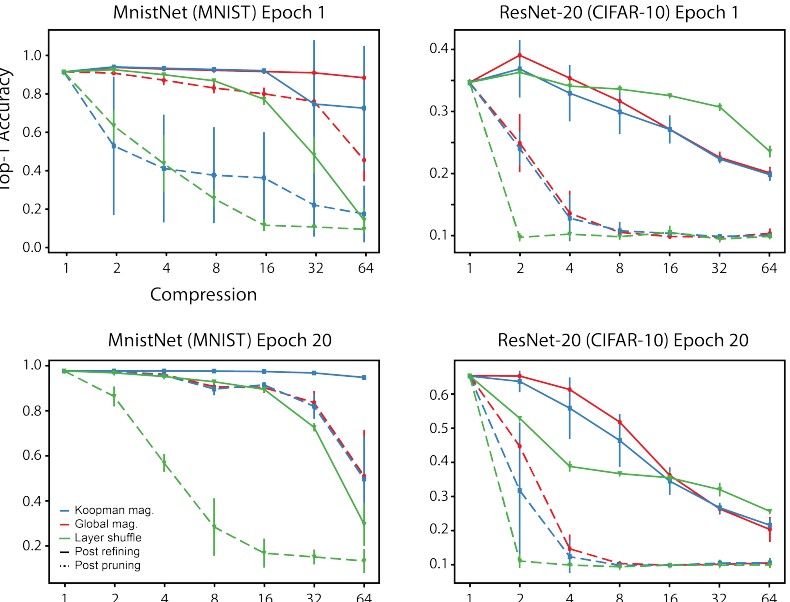

Figure 2: **Koopman and global magnitude pruning pre-convergence.** Left column, compression vs. accuracy of MnistNet trained on MNIST. Right column, compression vs. accuracy of ResNet-20 trained on CIFAR-10. Top row, results of pruning DNN after 1 epoch of training. Bottom row, results of pruning DNN after 20 epochs of training. In both cases, only data from epoch 1 and epoch 20, respectively, are used to compute the Koopman mode decomposition. Dashed lines are mean accuracy immediately after pruning and solid lines are mean accuracy after a single epoch of refining. Error bars are standard deviation.

## 4 KOOPMAN MAGNITUDE PRUNING PRE-CONVERGENCE

Motivated by the equivalence of Koopman and global magnitude pruning in the long training time limit, we next examined how the two methods compared before training converged. In this regime, we might expect KMP to outperform GMP, since computing the Koopman mode decomposition requires multiple snapshots of the training data. This provides KMP with considerably more dynamic information than GMP, which is based on a single snapshot.

As previous work has found that ResNet-20 trained on CIFAR-10 does not achieve linear mode connectivity until at least 5 epochs of training (Frankle et al., 2020), we examined pruning performance at epoch 1 and epoch 20 to capture the effect that linear mode connectivity had. We find that, despite the additional information, KMP performs similarly to GMP at both epochs (Fig. 2, right column).

There are two possible explanations for this similarity in performance. The first is that KMP and GMP prune the same parameters, making them equivalent methods (even before convergence). The second is that KMP and GMP prune distinct parameters, but those that remain result in a subnetwork with similar accuracy. To test these two hypotheses, we compare the overlap between the parameters not pruned by GMP and KMP at different compressions and different epochs (see Appendix B for more details on this metric). The right column of Fig. 3 shows that the two methods largely prune the same parameters. From epochs 1-10, the mean overlap is roughly $> 95\%$, even for a compression of 64. At epoch 20, there is a decrease in mean overlap and an increase in standard deviation. This suggests that the two methods are equivalent at the earliest epochs of training, with the equivalence breaking at some subsequent epoch.

Because KMP makes use of dynamical systems theory, the equivalence of KMP and GMP during (at least) the first 10 epochs has a strong implication on the trajectory in parameter space that is induced by training: at the end of each epoch, the largest parameters of the DNN must be the *same* ones that are expected to be the largest at convergence, if the order of training data were to remain the same. In this sense, the expected identity of the largest parameters are "static". The overlap between

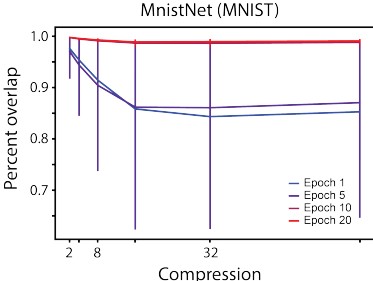 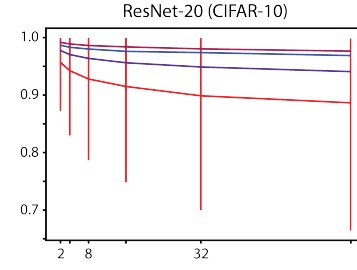

Figure 3: **Overlap between Koopman and global magnitude pruning masks.** Left, overlap of surviving parameters found via KMP and GMP on MnistNet training on MNIST. Right, overlap of surviving parameters found via KMP and GMP on ResNet-20 trained on CIFAR-10. Error bars are minimum and maximum of three seeds.

masks found using KMP from increasingly distant epochs shows a decrease with time (Appendix B, Fig. 5), suggesting that SGD slowly re-orders the identity of the largest parameters epoch-to-epoch, making these initial largest parameters inaccurate.

By epoch 20, KMP and GMP stop being equivalent and, on average, start pruning a larger number of different parameters. This implies that, even if the training data was ordered the exact same way for all the remaining epochs, the identity of (some of) the largest parameters are expected to evolve. To contrast this with the "static" dynamics found during epochs 1–10, we call this regime "active". We speculated that this change from "static" to "active" might be caused by emerging structure in the parameter evolution. To test this, we pruned ResNet-20 at epochs 1 and 20 using layer shuffle pruning (LSP). This method preserves the amount of compression in each layer, but randomly chooses which parameters to remove. Previous work has found that many state-of-the-art methods designed to prune at initialization do not do better than LSP (Frankle et al., 2021). We find that after only a single epoch of training, both GMP and KMP do no better, and in fact, do worse, than LSP for nearly all compressions tested. However, by epoch 20, both GMP and KMP were better than LSP for compression values of less than 16. These results support the idea that non-trivial structure is emerging in the DNN at a similar time that KMP and GMP stop being equivalent. When this structure is not present (e.g. at epoch 1), the two methods prune the same parameters.

We also examined the ability of KMP and GMP to prune MnistNet, trained on MNIST, after epochs 1 and 20. While we found similar results to ResNet-20 (Figs. 2 and 3, left column), there are subtle differences in the behavior of the overlap between pruning masks. For MnistNet, the overlap is smallest at epochs 1 and 5, suggesting that the two methods are not equivalent at the earliest part of training (Fig. 3, left). That this is different to what we found for ResNet-20 may be due to the fact that the two models are at different phases of training at equivalent epochs. Previous work has shown that, shortly after initialization, linear mode connectivity emerges in LeNet, trained on MNIST (Frankle et al., 2020). This does not happen for ResNet-20, trained on CIFAR-10, until at least epoch 5. Additionally, we find that, for MnistNet, KMP and GMP outperform LSP for compression above 8 as early as epoch 1. These MnistNet results further support the fact that, as structure beyond what LSP can find emerges, the dynamics associated with the identity of the largest parameters goes from being "static" to "active".

Finding what is driving this "static" to "active" transition, and whether it is related to the emergence of linear mode connectivity and/or other noted transitions during training (Gur-Ari et al., 2018; Achille et al., 2019; Kalimeris et al., 2019), will be the focus of future work.

## 5 "NEAR" EQUIVALENCE OF GLOBAL AND KOOPMAN GRADIENT PRUNING

Thus far, we have focused only on magnitude based pruning. In this section, we extend our work to another method that has found success and is of general interest: gradient based pruning (Yu et al., 2018; Lee et al., 2019; 2020; Blalock et al., 2020). In practice, this way of pruning is accomplished by scoring based on the gradient of the loss function, $\nabla_\theta L$, as propagated through the network. For instance, global gradient pruning (GGP) scores using the magnitude of the gradient times the

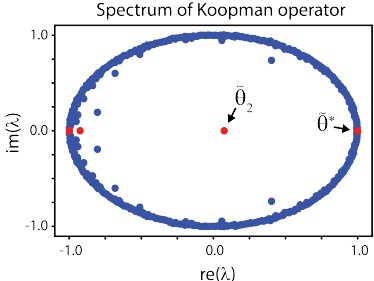 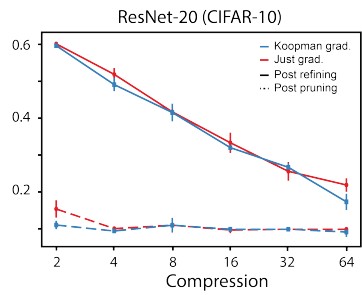

Figure 4: **Koopman gradient pruning.** Left, real and imaginary parts of the computed Koopman eigenvalues. Red dots denote modes that had a norm of greater than 0.85. Here, $\lambda_1 = 1.000$ and $\lambda_2 = 0.075$. Right, compression vs. accuracy of ResNet-20 trained for 20 epochs on CIFAR-10. Dashed lines are mean accuracy immediately after pruning and solid lines are mean accuracy after a single epoch of refining. Error bars are standard deviation.

parameter values, $s(\theta) = \left| \nabla_\theta L \cdot \theta(t) \right|$ (Blalock et al., 2020). Given that the gradient is applied to update the parameters of the DNN, $\nabla_\theta L$ is directly related to the parameters' dynamics.

From the Koopman mode decomposition, we have that the parameters at training step $t$, $\theta(t)$, are given by

$$\theta(t) = \tilde{\theta}^* + \sum_{i=2}^{R} \lambda_i^t \tilde{\theta}_i, \tag{5}$$

where the $\tilde{\theta}_i$ are the scaled Koopman modes (i.e. $\tilde{\theta}_i = \phi_i(p)\mathbf{v}_i$). As before, we use $\tilde{\theta}^*$ to denote the scaled Koopman mode that has $\lambda = 1$. Importantly, Eq. 5 tells us that the future state of the parameters is given by the sum of $\tilde{\theta}_i$ times their decaying eigenvalues. While there are $R - 1$ modes in this sum, those with large norms (i.e. $|\tilde{\theta}_i| \gg 0$) are expected to make the most substantial contribution. For simplicity, we consider only the largest modes that also have real, positive eigenvalues. These modes have easily interpretable dynamics (i.e. exponential decay) and are directly related to $\nabla_\theta L$. Pruning using the magnitude of these modes is what we call Koopman gradient pruning (KGP).

The Koopman spectrum of a ResNet-20 trained on CIFAR-10 for 20 epochs is shown in Fig. 4. In addition to the presence of an eigenvalue close to 1, corresponding to $\tilde{\theta}^*$, there is also an eigenvalue that satisfies the above criteria. We refer to its corresponding mode as $\tilde{\theta}_2$.

As expected from the discussion above, pruning ResNet-20 after 20 epochs of training using KGP is very similar to pruning using the scoring function $s(\theta) = |\nabla_\theta L|$, a method we refer to as just gradient pruning (JGP) (Fig. 4, right). We note that we only expect the two methods to be "nearly" equivalent (and not truly equivalent, as in the case of KMP and GMP in the long training time limit) because, KGP finds exponential decaying modes from dynamic data, whereas JGP is the average gradient computed over a batch of input data (Blalock et al., 2020).

These results highlight the fact that, while gradient and magnitude based pruning methods have seemingly little to do with each other, they are in fact, related in the Koopman operator framework. Whereas magnitude pruning is, at certain phases of training, equivalent to pruning using $|\tilde{\theta}^*|$, gradient pruning is equivalent to pruning using the exponentially decaying modes.

## 6 DISCUSSION

Inspired by the widespread success Koopman operator theory has found in a number of applied fields, and its recent application to problems in machine learning, we extended Koopman tools to study deep neural network pruning (Janowsky, 1989; Mozer & Smolensky, 1989a;b; LeCun et al., 1989; Karnin, 1990; Han et al., 2015; Blalock et al., 2020). By making use of the Koopman mode decomposition, we defined a new class of pruning algorithms (Algorithm 1) that are theoretically motivated and data-driven in implementation. We found that both global magnitude and gradient pruning (Blalock et al., 2020) have equivalent (or nearly equivalent) methods at certain regimes of

training in the Koopman framework (Figs. 1, 4). In this way, Koopman pruning provides a unified perspective on these "disparate" methods.

Applying KMP to DNNs in the early part of training revealed that it could be equivalent to GMP (Fig. 3). From a dynamical systems perspective, this equivalence implies that the largest parameters, at the end of each epoch, are the same ones that are expected to remain the largest, if training continued using the same ordering of the data. We refer to this behavior as "static". Given that we found the identity of the largest parameters evolved with training (Appendix B, Fig. 5), pruning too early may confine the sparsified DNN to a subspace where it stays in the static regime and therefore is unable find a good local minimum. This is additionally supported by the fact that layer shuffle pruning (LSP) outperforms GMP during this part of training, suggesting that the structure present in the GMP mask is more "harmful" than a randomly structured mask (Fig. 2).

However, at a certain subsequent point in training, the masks produced by GMP and KMP stop overlapping as completely (Fig. 3). This suggests that there is some transition in the dynamics, with the largest parameters becoming "active". As this transition seems to occur at a similar time that GMP starts to outperform LSP, and at a similar time that linear mode connectivity has been reported to emerge (Frankle et al., 2020), we speculate that it is related to the sparsified DNN still being able to evolve and find a good local minima in the subspace it is restricted to.

By making use of ShrinkBench, our results can be easily replicated and avoid some of the bad practices that have been recently identified as being pervasive in the DNN pruning literature (Blalock et al., 2020). Our custom implementations are integrated into the ShrinkBench framework, allowing for additional investigation into Koopman pruning.

While dynamical systems theory is a natural language with which to frame DNN optimization, the complex dependencies on optimizer, architecture, activation function, and training data have historically kept efforts in this direction to a minimum. This has led to a reliance on heuristic methods, such as iterative magnitude pruning, whose basis of success is still not clear (Elesedy et al., 2021). Other groups have attempted to more principally examine DNN behavior by studying spectral objects, such as the spectrum of the Hessian matrix (Gur-Ari et al., 2018) and the spectrum of the weight matrices (Martin et al., 2021). This work has been influential and led to new insight, however the connection between these metrics and the (future) dynamics of DNN training is not direct. An exception to this is the recent work on using inertial manifold theory to study the Lottery Ticket Hypothesis, which is similar in spirit to our Koopman based pruning methods (Zhang et al., 2021).

Given the data-driven tools that have been developed for Koopman operator theory (Mezić & Banaszuk, 2004; Mezić, 2005; Rowley et al., 2009; Tu et al., 2014; Williams et al., 2015; Mezić, 2020) and the rich theoretical literature that connects Koopman objects to the state space geometry (Mezić, 2005; Arbabi & Mezić, 2017; Mauroy et al., 2013; Mezić, 2020), recent work has focused on bring Koopman operator theory to the study of DNNs (Dietrich et al., 2020; Dogra & Redman, 2020; Tano et al., 2020; Manojlović et al., 2020; Mohr et al., 2021; Naiman & Azencot, 2021). This work has shown that Koopman methods can:

- Approximate the parameter dynamics during training and lead to speed-ups in wall clock time when substituted for standard optimizers (Dogra & Redman, 2020; Tano et al., 2020);

- Predict the number of layers necessary for a hierarchical support vector machine to achieve a given level of performance (Manojlović et al., 2020; Mohr et al., 2021);

- Shed light on the behavior of sequence neural models (Naiman & Azencot, 2021).

Our current work further illustrates the power that Koopman operator theory can bring. In particular, we found that the Koopman mode decomposition encodes dynamical information about training that could be leveraged to prune DNNs, enables a framework that unifies disparate pruning algorithms, and provides insight on how training dynamics impact the success of magnitude pruning. To the best of our knowledge, these could not have been done using any other linearization method (e.g. principal component analysis). We believe that continued exchange between the Koopman and machine learning communities will lead to further advances.

ACKNOWLEDGMENTS

W.T.R. was partially supported by a UC Chancellor's Fellowship. W.T.R., M.F., R.M and I.M were partially supported by the Air Force Office of Scientific Research project FA9550-17-C-0012. I. G. K. was partially supported by U.S. DOE (ASCR) and DARPA ATLAS programs.

REPRODUCIBILITY STATEMENT

Forethought to ensure reproducibility was taken in both the design and description of our methods and results. We made use of ShrinkBench, a recent open-source framework for developing and testing new pruning methods (Blalock et al., 2020), to implement all our experiments. This not only allowed us to report comparisons between our methods and existing methods across a number of different compressions, but will also enable easy sharing of our algorithms, as they are integrated into the ShrinkBench framework. Additionally, we have provided general pseudo-code for implementing Koopman based pruning (Algorithm 1) and a description of the method we used to compute the Koopman mode decomposition (Appendix A), to make it straightforward for others to understand and implement.

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

## A   PERFORMING KOOPMAN MODE DECOMPOSITION

As discussed in the main text, many different methods have been developed to perform Koopman mode decomposition. One very popular class of approaches is dynamic mode decomposition (DMD) (Schmid, 2010), and its many related variants. Seminal work by Rowley et al. (2009) showed that DMD, as originally proposed, could be equivalent to Koopman mode decomposition. Because of its popularity and noted connection to Koopman operator theory, we made use of one such DMD method, called Exact DMD (Tu et al., 2014), in this work.

Let $\mathbf{X} = [\theta(0), \mathbf{T}\theta(0), ..., \mathbf{T}^{(\tau-1)}\theta(0)] = [\theta(0), \theta(1), ..., \theta(\tau-1)] \in \mathbb{R}^{N \times \tau}$ be $\tau$ snapshots of a DNN's parameters, $\theta \in \mathbb{R}^N$, during all but the last iteration of training during an epoch (i.e. there are $\tau + 1$ iterations during an epoch). Here $\mathbf{T}$ is the nonlinear dynamical map that evolves the DNN parameters, which is dependent on the choice of optimization algorithm, architecture, activation function, and ordering of training data. Let $\mathbf{Y} = [\theta(1), \theta(2), ..., \theta(\tau)]$. Exact DMD does not require $\mathbf{X}$ and $\mathbf{Y}$ to be built from sequential time-series data, but we assumed it here.

Exact DMD considers the operator $\mathbf{A}$, defined as

$$\mathbf{A} = \mathbf{Y}\mathbf{X}^+, \tag{6}$$

where $^+$ is the Moore-Penrose pseudoinverse. Eq. 6 is the least-squares solution to $\mathbf{A}\mathbf{X} = \mathbf{Y}$, and hence, satisfies

$$\min_{\mathbf{A}} \left|\left| \mathbf{Y} - \mathbf{A}\mathbf{X} \right|\right|_F, \tag{7}$$

where $||\cdot||_F$ is the Frobenius norm. If $\mathbf{Y}\mathbf{c} = \mathbf{0}$ for all $\mathbf{c} \in \mathbb{R}^\tau$ such that $\mathbf{X}\mathbf{c} = \mathbf{0}$, then $\mathbf{A}\mathbf{X} = \mathbf{Y}$ exactly (Tu et al., 2014). This property is called linear consistency.

In practice, the DMD modes are found by computing the (reduced) singular value decomposition (SVD) of $\mathbf{X}$ (i.e. $\mathbf{X} = \mathbf{Q}\mathbf{\Sigma}\mathbf{V}^*$) and setting $\tilde{\mathbf{A}} = \mathbf{Q}^*\mathbf{Y}\mathbf{V}\mathbf{\Sigma}^{-1}$ (see Algorithm 2 and Appendix 1 of Tu et al. (2014) for more details on how the modes are computed from $\tilde{\mathbf{A}}$). When $\mathbf{X}$ and $\mathbf{Y}$ are linearly consistent and $\mathbf{A}$ has a full set of eigenvectors, the DMD modes are equivalent to the Koopman modes (Tu et al., 2014).

All pruning experiments performed and reported in the main text were done on a 2014 MacBook Air (1.4 GHz Intel Core i5) running ShrinkBench (Blalock et al., 2020). Computing the largest Koopman mode, $\tilde{\theta}^*$, took 44 seconds and 1.1 GB of memory usage for the ResNet-20 experiments, and 210 seconds and 1.1 GB of memory usage for MnistNet experiments. Memory usage was computed using the Python module memory-profiler 0.58.0[4]. A single epoch's worth of data (i.e. 391 iterations for ResNet-20 and 469 iterations for MnistNet) were used to construct the Koopman operator. Despite MnistNet having $\approx 1.6\times$ as many parameters as ResNet-20, the similar amount of memory usage required for both comes from the fact that we employed reduced SVD, which keeps only as many singular values as there are snapshots, which is comparable across the two DNNs (391 and 469).

### A.1   SCALING UP KOOPMAN PRUNING

Given that Koopman based pruning requires performing decompositions on matrices whose total number of elements scale as the square of the number of parameters present in the DNN, scaling up Koopman pruning to larger DNNs may seem intractable. However, the field of fluid dynamics, where Koopman has been extensively applied (Mezić, 2013), often deals with a similar problem, having the dimensionality of the state space being very large [e.g. $\approx 10^{5-9}$ (Luchtenburg & Rowley, 2011)]. Because of this, numerical methods have been developed, such as the method of snapshots (Sirovich, 1987) to efficiently compute the SVD of large matrices. These can, and have, been easily incorporated into methods used to compute the Koopman mode decomposition. Therefore, we expect that leveraging similar approaches should make scaling Koopman pruning to larger DNNs tractable.

---

[4]https://pypi.org/project/memory-profiler/

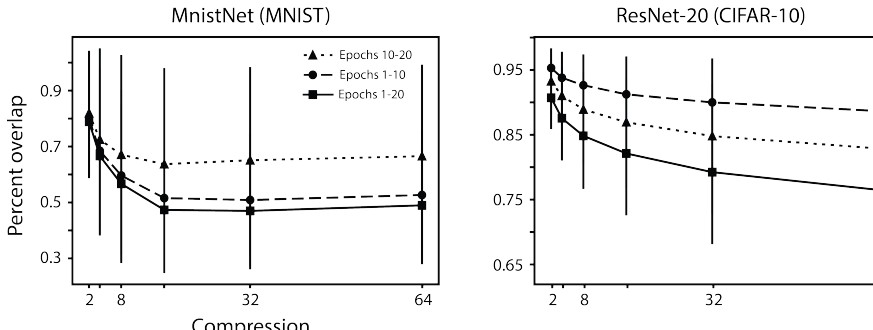

Figure 5: **Overlap between Koopman masks across epochs.** Left, overlap of surviving parameters found via KMP on MnistNet, trained on MNIST, at different epochs. Right, overlap of surviving parameters found via KMP on ResNet-20, trained on CIFAR-10, at different epochs. Error bars are standard deviation.

## B  PRUNING MASK OVERLAP

Just because two pruning methods find subnetworks that have the same performance at a given compression does not imply that the two methods prune the same parameters. Therefore, to study how similar two approaches, $A$ and $B$, are at the level of individual parameters, we computed the pruning overlap of the two masks, $m^A$ and $m^B$, defined as

$$o = \sum_i \left( m_i^A \cdot m_i^B \right) \Big/ (N/c). \tag{8}$$

Note that the denominator is equivalent to the number of non-pruned parameters, and that, since $m \in \{0,1\}^N$, $m_i^A \cdot m_i^B = 1$ if and only if both masks agree to save parameter $i$.

While we explored the overlap of Koopman magnitude pruning and global magnitude pruning masks in the main text, we can also use the overlap metric to compare how masks found using the same method changes when applied at different points during training time. In this case, $m^A$ corresponds to the mask found using a given method at epoch $A$ (and similarly for $B$). We did this analysis for KMP masks, finding that for both MnistNet and ResNet-20, there is change in which parameters are kept across training time (Fig. 5).

