# OpenReview forum: "An Operator Theoretic View On Pruning Deep Neural Networks"
_ICLR.cc/2022/Conference — ICLR 2022 Poster_

### Official Review · Reviewer_zqcK · 2021-10-27

**Correctness:** 3
**Technical Novelty And Significance:** 2
**Empirical Novelty And Significance:** 2
**Recommendation:** 6
**Confidence:** 4

**Main Review:**

The paper is overall well written. There were several aspects of this paper that I liked a lot:

- I think that the questions asked in the paper, e.g., why has magnitude-based pruning been successful and what is its relationship with gradient-based pruning, are highly sought-after.

- I also appreciate the efforts that the authors made to ensure the reproducibility of the paper as mentioned in the discussion section.

Having said this, at a few places in the paper, I am slightly confused by the authors' interpretation of their results and feel that they might be somewhat overstated. Two such places a below -- I am happy to be corrected if wrong:

- The first place is Section 3, where the authors state that the global and magnitude pruning in the long training time limit is *equivalent*. Equivalence is a strong claim and seemingly this is unlike to hold without strong assumptions on the dynamics of neural network parameters. Indeed, to my understanding, the Koopman operator approximates the dynamic of neural network parameters using a linear transformation, as in Appendix A. This seems to suggest that (4) only holds in a very special regime of neural network training, where the parameter updates can be modeled by a linear update matrix. The assumption of "linear dynamics in parameters'', however, seems to be a big constraint -- it is known that even linear neural networks (without any nonlinear activation) have nonlinear dynamics in parameters [(Saxe et al. 2013)](https://arxiv.org/abs/1312.6120). To clarify this point, I suggest the authors either (i) state a theorem on the equivalence with all the necessary assumptions listed or (ii) slightly tone down the equivalence claim.

- The second place that I got a bit confused about is how exactly does Koopman pruning provide "new insight on the success (or lack thereof) of magnitude pruning pre-convergence," which is one of the main contributions listed in the introduction. I believe that Section 4 was designated to explicate this insight, but I feel that there is no strong message in that section that I can take home. I feel that the text and experiments in Section 4 demonstrated that Koopman pruning and magnitude-based pruning are *not* equivalent in pre-convergence (by epoch 20 or so) -- but is that all the authors want to conclude for magnitude pruning at pre-convergence?

**Questions**:

- In Section 2.1, where the authors cast neural network training into Koopman decomposition, it will be great if the authors can explicitly write the interpretation of $\mathbf{U}$, $g$, and $\mathbf{T}$ in the context of neural network training. Based on my understanding, in neural network training, $g$ is the identity function, $\mathbf{T}$ is the parameter-update transformation modeled by a linear operator, and $\mathbf{U}$ is identical to $\mathbf{T}$. Is this interpretation correct? If so, perhaps it is worth mentioning already at this point that throughout this paper, we assume that the neural network update dynamics is modeled by a linear operator (a matrix) $\mathbf{T}$.

- In Section 2.2. and Algorithm 1, the authors assume that $\lambda_1 = 1$ and refer this to as "a reasonable assumption in the case of stable dynamical systems". I am wondering if this assumption is well-corroborated by experiments. That is, when we perform Koopman decomposition on snapshots of parameters, is $\lambda_1$ close to 1? I am not sure if I would expect it to be the case. I feel that the reason perhaps is not that the dynamical system of parameter evolution is "unstable'' but rather that it may not be well-approximated by a linear dynamical system.

**Summary Of The Paper:**

The authors studied network pruning from the perspective of dynamical system theory. They show that a new type of pruning method, named Koopman pruning, unifies magnitude pruning and gradient-based pruning to a degree. It also clarifies aspects of magnitude-based pruning before convergence in training.

**Summary Of The Review:**

To summarize, I feel that the paper can be a meaningful contribution to the field. However, the "operator theoretic perspective" of this paper -- to my feeling as a non-expert -- is somehow overstated. I think that Koopman pruning is an interesting and useful pruning heuristic on its own, but I am not sure if it can genuinely explain magnitude-based pruning either before or after training convergence in a realistic setting, where network parameters cannot be modeled by a linear dynamical system.

---

> ### Author Response · Authors · 2021-11-20
> **Response to Reviewer zqcK**
>
> A. This is a good point, especially given that some of the (future) readers may not have strong expertise in Koopman operator theory or dynamical systems.
>
> It is known in the Koopman community that the infinite dimensional Koopman operator, which is linear, is able to fully encode even nonlinear dynamics. However, we cannot construct or work with infinite dimensional operators. Subsequent work has shown that finite dimensional approximations of the Koopman operator can capture important features of nonlinear dynamical systems, but certain conditions on the Koopman eigenfunctions have to be met. While sometimes challenging to show in practice, these theoretical results are part of why Koopman tools have been so broadly applied to study many complicated, nonlinear dynamical systems.
>
> Beyond these Koopman specific results, foundational work in dynamical systems theory led to the Hartman-Grobman theorem, which states that a nonlinear system can be linearized in a neighborhood of a hyperbolic fixed point, with the linear and nonlinear systems being conjugate. Subsequent work by Lan & Mezić, 2013 extended this theorem to the entirety of the basin of attraction. This means that, as long as the parameters of the DNN under consideration are in the basin of attraction of a local minimum, we can find a linearization that properly captures the relevant dynamical features. Therefore, we can assume that global magnitude pruning (GMP) and Koopman magnitude pruning (KMP) will indeed be equivalent in the long training time limit, where we assume that the DNN is in a basin of attraction. These references and the necessary assumptions have now been added to Secs. 2 and 3.
>
> B. The unexpected result that KMP and GMP are equivalent early in training (for ResNet-20 trained on CIFAR-10) led us to utilize the dynamical systems perspective of Koopman operator theory to make the following observation. The equivalence implies that the largest parameters, at the end of each epoch, are the same ones that are expected to stay the largest, if training continued using the same ordering of the data. We refer to this behavior as “static”. Given that we found the identity of the largest parameters evolved with training (Appendix B, Fig. 5), pruning too early may confine the sparsified DNN to a subspace where it stays in the static regime and is, therefore, unable find a good local minima. This is additionally supported by the fact that layer shuffle pruning (LSP) outperforms GMP during this part of training, suggesting that the structure present in the GMP mask is more “harmful” than a randomly structured mask (Fig. 2).
>
> However, at a certain subsequent point in training, the masks produced by GMP and KMP stop overlapping as completely (Fig. 3). This suggests that there is some transition in the dynamics, with the largest parameters becoming “active”. As this transition seems to occur at a similar time that GMP starts to outperform LSP, and at a similar time that linear mode connectivity has been reported to emerge (Frankle et al., 2020), we speculate that it is related to the sparsified DNN still being able to evolve and find a good local minima in the subspace it is restricted to.
>
> We agree that these interpretations of the experiments were not as clearly laid out in our original submission and have revised our discussion in Sec. 6 to make these points more apparent. We additionally changed our wording of these results from "new insight on the success (or lack thereof) of magnitude pruning pre-convergence" to "new insight on how the training dynamics may impact the success (or lack thereof) of magnitude pruning pre-convergence” to convey a more accurate representation of what is supported by our experiments.
>
> C. We have added the interpretation of g and T in the context of neural network training, to Sec. 2.1. Note, however, that U and T are not equivalent, because Eq. 1 holds for all possible observables in the considered function space.  As mentioned in the first point, we hope that the addition of the discussion around the appropriateness of considering a linearization inside a basin of attraction will help make it clear that we are considering the neural network update dynamics to be modeled by a linear operator, but still capturing the same information present in the full, nonlinear system.
>
> D. The assumption of $\lambda_1$ being equal to 1 was well corroborated by our experiments. As can be see in the left subpanel of Fig. 4, the red dot corresponding to the eigenvalue of $\tilde{\theta}^*$ is almost exactly 1. All other experiments found similar results. This point has been added to Sec. 2.2 and the value of the $\lambda_1$ and $\lambda_2$ have been added to the caption of Fig. 4.

---

> > ### Comment · Reviewer_zqcK · 2021-11-26
> > **Response to the authors' rebuttal**
> >
> > Thank you for your response. Especially, I find your responses C and D very helpful and addressed my questions. However, after reading A and B and the paper multiple times, it is still unclear to me if the claims:
> >
> > (1) global magnitude pruning (GMP) and Koopman magnitude pruning (KMP) are equivalent in the long training time limit
> >
> > (2) KMP and GMP are equivalent early in training
> >
> > are heuristic arguments or if the authors claim they are mathematically true. More specifically, I am wondering if one can show theorems saying that KMP and GMP pick out the same parameters to prune -- Is this true and does it make sense? If this is what the authors intend to claim, I suggest writing out such theorems (with all necessary assumptions that the authors find reasonable to impose), instead of just colloquially describing them in passing.

---

> > > ### Author Response · Authors · 2021-11-26
> > > **The magnitude pruning equivalence claims**
> > >
> > > We are glad our responses were helpful and we appreciate the additional feedback.
> > >
> > > The first equivalence (long training time limit) is mathematically true, given the assumptions outlined in the paper (i.e. that the training is in a basin of attraction of some local minimum and that the DNN parameters are sufficiently “close” - with respect to the $l^2$ norm - to that minimum). This makes sense because the first Koopman mode (i.e. $\tilde{\theta}^*$) will well approximate the true local minimum, and thus, will also well approximate the current state of the DNN (see Eq. 4). We therefore expect KMP and GMP will prune the same parameters. We agree that writing this out in the format of a theorem would be helpful, and will add that our revised manuscript.
> > >
> > > The statement of the second equivalence (early training) was motivated and supported by experimental evidence (e.g. the overlap of the parameters not pruned by GMP and KMP being >90-95%). This was then used to make a statement about the dynamics present during the early part of training. However, a theorem can be made that states that, if a given DNN has its largest magnitude parameters grow at the largest rate over some sufficiently long interval of training, then KMP and GMP will be mathematically equivalent. This again makes sense, as the dynamical information that KMP has access to will predict that the largest parameters will continue to be the largest at the local minimum the DNN converges to. Therefore, the two methods will again prune the same parameters. We believe that stating this theorem will be helpful for understanding the implications of Sec. 4, and may also be a point of interest. It will therefore be added to our revised manuscript.

---

> > > > ### Comment · Reviewer_zqcK · 2021-12-01
> > > > **Response to the authors' rebuttal**
> > > >
> > > > Thank you for clarifying both equivalences. I've raised my score.

---

> ### Comment · Area_Chair_cmfV · 2021-11-26
> **Please respond to the author rebuttal**
>
> Dear Reviewer zqcK,
>
> The authors have posted their rebuttal. I wonder whether the rebuttal addressed your concerns? Please respond to the authors. Thanks!
>
> AC

---

### Official Review · Reviewer_Q2k5 · 2021-11-02

**Correctness:** 4
**Technical Novelty And Significance:** 2
**Empirical Novelty And Significance:** 2
**Recommendation:** 6
**Confidence:** 3

**Main Review:**

Strength:
1. The paper is clearly written.
2. The experiments are motivated and executed well.
3. Provides a theoretical basis for pruning which is an intuitive and successful procedure.
4. The difference between the static and active regimes is interesting. While why this happens is not answered in the paper, it can be considered as a useful by product.

Weakness:

1. Is the message tautological? As in, DNN training is a hidden procedure and the Koopman operator again projects onto a high dimensional space so as to linearise the dynamics, and once this is done, it is sort of natural to expect that the top eigenvalue will have the most say. Now, the Koopman operator is also sort of hidden and does not provide further insights into the working of the DNN. Would any sort of linearisation of dynamics not lead to similar conclusions.

2. While Algorithm 1 is a "general form", it is only conceptual, in that, it does not directly yield a procedure that speeds up "wall clock time".





**Summary Of The Paper:**

The paper uses a Koopman operator to provide a theoretical basis to explain the success of pruning deep neural networks.

**Summary Of The Review:**

The paper clearly presents interesting results, the experiments seem sound. However, the novelty/significance is not overwhelming and hence the borderline score.

---

> ### Author Response · Authors · 2021-11-20
> **Response to Reviewer Q2k5**
>
> A. We appreciate the point you are raising, and indeed agree that, a priori, it is not clear whether lifting to an infinite dimensional space would bring any insight into the workings of a given DNN. However, the findings of this paper show that some features identified by the Koopman mode decomposition are analogous to quantities that the machine learning community is commonly interested in (e.g. the local minimum the DNN converges to, the gradients of all the parameters). To the best of our knowledge, such analogies do not exist for other linearization methods (e.g. principal component analysis). This difference between Koopman and other methods comes, in part, from the fact that the Koopman mode decomposition is directly related to the (future) dynamics of the system, and is therefore interpretable.
>
> We have added additional discussion to make this point clearer in the revised manuscript’s Sec. 6.
>
> B. We agree that Algorithm 1 is conceptual and does not directly yield a procedure to speed up wall clock time. We do not believe we suggested that it did, only that prior work on applying Koopman operator theory to accelerate training exists (Dogra & Redman, 2020; Tano et al., 2020). Such prior work employed additional considerations not presented in Algorithm 1.

---

> > ### Comment · Reviewer_Q2k5 · 2021-11-29
> > **Thanks for the response**
> >
> > The authors do seem to agree on my major comments, and hence my assessment of the paper remains unchanged.

---

> > > ### Author Response · Authors · 2021-11-29
> > > **Response to Reviewer Q2k5**
> > >
> > > Thank you Reviewer Q2k5 for your response. We politely point out that, in our rebuttal, we agreed that your first major comment was an important one to address, but we did not agree with its conclusion (i.e. that the "message [of the paper] is tautological"). In particular, we discussed how our results illustrate why this is not the case, as the Koopman spectral quantities are found to be equivalent to quantities the ML community is interested in (e.g. local minimum, gradients). In addition, our rebuttal agreed that Algorithm 1 did not yield a reduction in wall clock time, but this was never claimed to be the case. We therefore do not believe it is a relevant major critique of our paper.

---

> ### Comment · Area_Chair_cmfV · 2021-11-26
> **Please respond to the author rebuttal**
>
> Dear Reviewer Q2k5,
>
> The authors have posted their rebuttal. I wonder whether the rebuttal addressed your concerns? Please respond to the authors. Thanks!
>
> AC

---

### Official Review · Reviewer_Zwfq · 2021-11-02

**Correctness:** 4
**Technical Novelty And Significance:** 3
**Empirical Novelty And Significance:** 3
**Recommendation:** 6
**Confidence:** 3

**Main Review:**

This paper tries to explain the success of magnitude based and gradient based methods in compressing neural network models. Although these methods are naive, both of them are found to be robust and are easy to implement.

strengths:
1. This paper is well-written and easy to follow.
2. The targeting problem of this work is very interesting and fundamental.
3. The solution proposed in this work is easy to implement and is supported by solid theory. For practitioners who are familiar with eigen-decomposition, the theory part of this work may not be amazing, however this viewpoint on understanding pruning is very interesting and provides valuable insights into the success of pruning based methods.


questions:
1. This works presents results on GMP. I am wondering if the proposed framework can be applied to layerwise pruning. Although in [1], the layerwise pruning is outperformed by GMP.
2. What are the  computational  requirements? For example, to compute the Koopman mode decomposition of ResNet-20 on CIFAR10, how much time is needed? How much memory is needed? How many snapshots are needed to obtain the eigenvector associated to the largest eigenvalue?
3. Is the proposed approach be scalable to large networks? It seems there are some computational challenges since the dimension of the parameter space of a large network can be very high.



[1] Blalock, Davis, Jose Javier Gonzalez Ortiz, Jonathan Frankle, and John Guttag. 2020. “What Is the State of Neural Network Pruning?” ArXiv:2003.03033 [Cs, Stat], March. http://arxiv.org/abs/2003.03033.


**Summary Of The Paper:**

This paper proposes a new class of pruning algorithms based on Koopman operator theory and shows existing magnitude based pruning methods and gradient based pruning methods can be unified under the proposed framework. This is a very interesting approach to justify the success of magnitude based and gradient based methods in compressing neural networks by utilizing eigendecomposion, which has been successfully applied to understand the structure of RKHS and GPs.

**Summary Of The Review:**

This paper provides an interesting viewpoint in understanding the success of pruning based compression methods. The method is easy to implement and is very practical  (at least for small-scale networks).

---

> ### Author Response · Authors · 2021-11-20
> **Response to Reviewer Zwfq**
>
> A. Koopman magnitude pruning (KMP) can indeed be extended to layerwise magnitude pruning in a straightforward manner. The only difference to global KMP is that the estimated first Koopman mode, $\tilde{\theta}^*$, would be split into subvectors, corresponding to the different layers. Each layer would then be compressed by a pre-determined amount, using a threshold computed on the magnitudes of that subvector.
>
> As you point out, Blalock et al., 2020 found that global magnitude pruning outperforms layerwise magnitude pruning, which is what led us not to study it here. However, we have added mention of this to the revised manuscript, to emphasize this additional connection between Koopman based pruning and existing pruning methods.
>
> B & C. The question of computational requirements is a pertinent point that we have now incorporated into the Sec. 2.1 and Appendix A.
>
> To obtain the Koopman eigenvector associated with the largest eigenvalue for ResNet-20, 44 seconds and 1.1 GB of memory usage were required. An epoch worth of data was used (i.e. 391 iterations). To obtain the Koopman eigenvector associated with the largest eigenvalue for MnistNet, 210 seconds and 1.1 GB of memory usage were required. An epoch worth of data was used (i.e. 469 iterations). Note that the memory usage is comparable between the two (despite the fact that MnistNet has approximately $1.6\times$ as many parameters as ResNet-20), because we used reduced SVD (i.e. we kept only as many singular values as there snapshots).
>
> Given that Koopman pruning requires performing decompositions on matrices whose total size scales as the number of DNN parameters squared, applying Koopman pruning to larger DNNs may seem intractable. However, the field of fluid dynamics, where Koopman operator theory has been extensively applied (e.g. Mezić, 2013), often deals with a similar problem. In particular, it is not uncommon for the dimensionality of the underlying state space to be very large (e.g. ~$10^{5-9}$) (Luchtenburg & Rowley, 2011). Because of this, numerical methods have been developed, such as the method of snapshots (Sirovich, 1987), to efficiently compute the SVD. These can be, and have been, easily incorporated into methods used to compute the Koopman mode decomposition. Therefore, we expect that leveraging similar approaches should make scaling Koopman pruning to larger DNNs tractable.

---

> > ### Comment · Reviewer_Zwfq · 2021-11-29
> > **score slightly decreased**
> >
> > I thank authors' response. From authors' responses, it is still unclear to me how to perform a layer wise compression using the proposed approach. The scaling ability (performance) is unclear either without  empirical results. I would like to emphasize I like the viewpoint  very much. More empirical experiments will make this work much more convincing.

---

> > > ### Author Response · Authors · 2021-11-29
> > > **Layer wise compression**
> > >
> > > Thank you Reviewer Zwfq for your response. We are happy to hear that you like the Koopman viewpoint.
> > >
> > > To clarify layer wise compression using Koopman pruning, we provide the following algorithm:
> > >
> > > ------------
> > > 1) Construct the data matrix $D = [\theta(0), \theta(1), ..., \theta(\tau)]$ from $\tau + 1$ snapshots of the parameter values $\theta$ during training.
> > > 2) Compute the Koopman mode decomposition from $D$. Identify the first Koopman mode, $\tilde{\theta}^*$. Note that $\tilde{\theta}^*$ has values associated with all the parameters in the DNN.
> > > 3) Break $\tilde{\theta}^*$ into subcomponents $\{\tilde{\theta}^*_1, ...,  \tilde{\theta}^*_M\}$, according to which of the M layers each parameter belongs to. For instance, if the first layer has 100 parameters, then $\tilde{\theta}^*_1$ will be made up of the first 100 values of $\tilde{\theta}^*$.
> > > 4) Score the parameters in the each layer by their magnitude in the first Koopman mode, using the function $s_i = |\tilde{\theta}^*_i|$, for $i = 1,..., M$.
> > > 5) Compress each layer by an amount $c$, removing (or "masking out") the parameters that are not in the top $1/c$ percentile of $s_i$.
> > > --------------
> > >
> > > The steps outlined above yielded a pruned DNN that has been compressed by an amount $c$ in each layer. We will add this algorithm to the Appendix, in order to make the possibility of layer wise compression using Koopman methods more clear.

---

> ### Comment · Area_Chair_cmfV · 2021-11-26
> **Please respond to the author rebuttal**
>
> Dear Reviewer Zwfq,
>
> The authors have posted their rebuttal. I wonder whether the rebuttal addressed your concerns or whether other reviewers' comments changed your evaluation? Please respond to the authors. Thanks!
>
> AC

---

### Official Review · Reviewer_kuYT · 2021-11-06

**Correctness:** 3
**Technical Novelty And Significance:** 3
**Empirical Novelty And Significance:** 3
**Recommendation:** 6
**Confidence:** 2

**Main Review:**

I guess the main strength of this paper is the perspective the authors use to look at this problem, which to me sounds new. Other merits of this paper is that it is readable and teh authors have made an effort to provide a tutorial on Koopman theory, depspite this subject being quite specialistic.

**Summary Of The Paper:**

This paper attempts to explain why magnitude-based and gradient-based pruning is effective only after the network has converged, ie it has learned the given task. The aothors attempt to shed some light on this issue via Koopman theory, borrowing from the field of dynamic systems.

**Summary Of The Review:**

While this paper has the merit of the originality of the approach, it is hard for me to provide a string feedback due to the unfamiliarity with this Koopman theory.

---

> ### Comment · Area_Chair_cmfV · 2021-11-12
> **Would you please enrich your review comment?**
>
> Dear Reviewer kuYT,
> Thanks for reviewing the paper! Unfortunately, your review is too terse. Would you please make your review better supported? I understand that you may not be a good fit for the paper. Thanks!
>
> AC

---

### Comment · Area_Chair_cmfV · 2021-11-12
**Discussion needed**

Dear Reviewers,
Thanks for reviewing the paper! However, there is some discrepancy in your ratings. Especially, reviewer zqcK is stricter than others. Would you please look at each other's review and post some opinions? If you deem your rating is OK, please try to defend a bit, rather than simply posting "I keep my score" only. Of course, if you would like to post after the authors submit their rebuttals, fine. Thanks!

AC

---

### Author Response · Authors · 2021-11-20
**General Response to all reviewers**

We thank the reviewers for their time and feedback.

From the reviews, it is clear that more detail on practical and theoretical considerations could enhance the rigor and quality of our work. We have therefore strived to address this in the revised manuscript. In particular, we have added discussion on:

1. Scaling Koopman pruning up to larger DNNs (R2);

2. Contrasting Koopman mode decomposition with other linearization methods (R3);

3. Using Koopman mode decomposition in regimes where the parameter dynamics are not linear (R4);

4. Interpreting the Koopman magnitude pruning results in the pre-convergence phase of training (R4).

More detail on each of these points, and additional questions that were raised, can be found in the responses to each reviewer and the revised manuscript.

---

### Comment · Area_Chair_cmfV · 2021-11-28
**Please post your post-rebuttal opinion!**

Dear Reviewers,
The authors have updated their manuscript and responded to your comments. Please check whether your concerns have been addressed and then post your further opinions *if you haven't*. This is the professional way to show respect to the authors' efforts. The deadline Nov. 29 is coming very soon. Thanks!

AC

---

### Decision · Program_Chairs · 2022-01-20

**Decision:**

Accept (Poster)

**Comment:**

The paper used the Koopman operator theory to explain and guide the DNN pruning. All the reviewers deemed that such a viewpoint is novel (but at different levels). However, the paper still had some issues, including unclear technical details, vague/overselling statements, being computation and memory expensive, etc. The paper finally got 4 "marginally above threshold" (one being of low confidence), making it on the borderline. The AC read through the paper and agreed that the Koopman operator theory brings new perspective to DNN pruning, with potential for other analysis of DNNs. Although the paper is imperfect and not strong, it does not have severe problems either and the issues pointed out by the reviewers could be easily fixed (except the scalability issue, which can be left as future work). In order to encourage new ideas, the AC recommended acceptance.